ecology, environmental science, computational biology

spatially explicit model, rewilding, seed dispersal, ecological function

**Author for correspondence:**
Hugo Thierry
e-mail: hugothierryp@gmail.com

# Where to rewild? A conceptual framework to spatially optimize ecological function

Hugo Thierry and Haldre Rogers

Department of Ecology, Evolution, and Organismal Biology, Iowa State University, 251 Bessey Hall, Ames, IA 50011, USA

(iD) HT, 0000-0002-7600-1160

Rewilding is an approach aiming at restoring ecosystems to a self-sustaining state by restoring ecological function through active reintroductions or passive management. Locations for most rewilding-through-reintroduction efforts today are selected based on the suitability of the habitat for the reintroduced species, often with little consideration of where the ecological function is most needed. We developed the Spatial Planning of Rewilding Effort (Spore) framework to identify priority locations for rewilding projects through simultaneous consideration of habitat suitability and provisioning of ecological function. We use the island of Guam as a case study for a potential rewilding project, as the island has functionally lost all native seed dispersers as a result of the invasive brown treesnake (*Boiga irregularis*). The Såli (Micronesian starling, *Aplonis opaca*) is a good candidate for rewilding to restore ecological function, because it is an effective seed disperser with a localized remnant population. Using Spore, we identify three priority areas for the restoration of seed dispersal, each subdivided into management units. By recognizing the influence of landscape structure and the behaviour of the reintroduced species on the spatial pattern of the function provided by that species, this approach should lead to improved ecological outcomes.

## 1. Introduction

While the term 'rewilding' has been used to describe a range of conservation actions in the last decades [1,2], there is general acceptance that the goal of rewilding is to restore ecosystems to a self-regulating state requiring minimal human intervention [3,4] but see Hayward *et al*. [5]. To allow for self-regulating systems, it is essential to restore key ecological functions that have been extirpated from the system. The active approach to rewilding is through the reintroduction of locally extinct, or in rare cases the translocation of ecologically similar species, to perform missing ecosystem functions [6,7], and restore functional complexity [8]. The IUCN guideline for reintroductions and other conservation translocations [9] is a resource for helping decide how and when to move plants and animals for conservation purposes.

Historically, rewilding efforts have focused on trophic rewilding: reintroducing large vertebrates, primarily carnivores extirpated anytime from the recent past to the Pleistocene, across large scales to restore trophic cascades [10]. Trophic rewilding of the grey wolf (*Canis lupus*) in Yellowstone National Park, for example, led to multiple indirect positive impacts on numerous ecological functions such as carbon storage [11], biodiversity enhancement [12], reestablishment of native plant diversity [13], riparian restoration [14] and regulation of diseases [15]. Projects aiming at restoring mutualistic interactions, such as seed dispersal and pollination fall under mutualistic rewilding [16,17]. In areas where species can recolonize without human intervention, minimal human interference allow passive rewilding. How interference and baseline setting influence rewilding are well described in Fernández *et al*. [8]. Rewilding is particularly relevant on islands due to the

high level of species extirpation and extinctions, and tends to focus on invasive eradication [18], reintroduction, and the use of taxon substitutes [19]. Regardless of the type of rewilding, areas for rewilding are often chosen opportunistically using criteria such as ownership [18] and accessibility [20]. Recent tools have been developed to monitor rewilding projects [21], and measure outcomes through time, but few tools exist to guide the selection of areas when initiating these projects.

While ecological function is often considered as spatially homogeneous across the range of the reintroduced species in these studies, factors such as behaviour [22] and landscape structure [23] lead to functional heterogeneity at fine spatial scales. A predator and a prey could share the same landscape, but differences in habitat usage or temporal niche could limit predation, and thus rewilding may not produce the desired ecological function. Failure to consider such potential mismatches can even lead to negative impacts through functional disservices [24]. Much is known about the spatial distribution of ecological functions, such as predation [25], pollination [26] or water quality [27]. Nevertheless, applications still rarely link animal reintroductions to fine-scale spatial patterns of ecological function.

Spatial models are a useful tool for merging the ecological goals of conservation reintroductions with rewilding. They allow for the exploration of interactions between biotic and abiotic processes under different theoretical scenarios [22]. Conservation biology and reintroduction have relied on habitat suitability approaches [28] and spatially explicit platforms such as Zonation [29], ATLAS [30] or MARXAN [31] to plan conservation actions. In parallel, models mapping the spatial patterns of ecological function have been developed, particularly in the field of ecosystem services. Spatially explicit models such as InVEST [32] map ecosystem services at fine spatial scales and thus highlight areas of interest in relation to management goals [33]. Individual-based models have also been used to explore how species provide functions across space [34] in relation to habitat requirements [35], dispersal [36], seasonality [37] and diet [38].

While all these models are available, few approaches have combined both habitat suitability and spatial distribution of ecological function in one framework. Sobral-Souza *et al.* [39] have taken a step in this direction, exploring the spatial pattern of ecological function provided through rewilding at a sub-continental scale. While such large scales could help in deciding which broad areas within a region to focus on, most rewilding attempts and relevant ecosystem management decisions are made at smaller spatial scales. Local stakeholders are used to proceeding at such fine scales and their involvement has been proven to have considerable impacts on the ecological success of management projects [40].

Here, we provide a framework for planning conservation translocation or range expansions for the purposes of restoring ecological function and illustrate the use of the model with a case study. The Spatial Planning of Rewilding Effort (Spore) framework supports rewilding projects that explore the spatial distribution of ecological functions provided at fine spatial scales as well as the habitat suitability for rewilding species. The ecological function needs and the suitable habitat for the function provider are combined to identify rewilding strategies that optimize the restoration of ecological function to move towards the goal of a self-regulating system. We also create management units to facilitate management actions [41].

## 2. Material and methods

### (a) The spore framework

The Spore framework is a spatially explicit, grid-based approach for prioritization of rewilding projects. Spore divides the landscape into a spatial grid with a user-defined resolution across which the model evaluates where the ecological function could be provided and the suitability of habitat for the function provider. The resulting layers are then combined, aggregated into management units and ranked according to the potential for restoration of ecological function. The scoring of ecological function is user-defined and can represent one or an ensemble of functions. Throughout this application, we focus on a single ecological function to avoid complexity and provide a straightforward case study. Nevertheless, the ecological function is often provided by a suite of species that each can also provide other key functions and selecting certain species to reintroduce and favouring a specific ecological function could lead to negative impacts on other species and ecological functions. The Spore framework is built to allow increasing complexity, by including multiple species and/or ecological functions, so a user could extend the Spore framework to identify complementarities and tradeoffs. The model works by assigning scores to each cell at each step of the framework, then combining cells with positive rewilding scores into management units (figure 1). The Spore framework can be applied using spatially explicit modelling platforms such as GAMA [42] and Netlogo [43] which are capable of handling geographic information system (GIS) data and grid-based entities using intuitive functions.

To illustrate our framework, we developed Spore-Guam, an application of Spore to a case study on the island of Guam. Spore-Guam is developed in the GAMA modelling and simulation development environment. We chose to develop our model in this platform because the GAML language used in GAMA allows easy manipulation of GIS data and spatial queries for a large number of entities. All scores and functions used in the model are described in detail in the electronic supplementary material, appendix A. All code used can be found online at the following address: https://github.com/EBL-Marianas/Spore.

#### (i) Conceptualization and data input

In preparation for the model, users must first identify the target ecological function to be restored, and the organisms that can provide this function. Knowledge of both entities is then used to conceptualize the spatial relationship between them to develop the algorithms used in the rest of the framework. Spore uses a GIS-based land cover map as a baselayer representation of the studied system.

Our case study focuses on rewilding native and degraded forest on the island of Guam through restoring avian frugivores that provide seed dispersal functions. Guam (13°27′ N, 144°46′ E; 543.9 km$^2$) is a US territory and the largest island in the Mariana archipelago, located in Micronesia. The dominant forest type of the island is karst forest, which previously covered the majority of the northern half of the island, but now remains on only about 12% of the island. During World War II and in the development since then, large tracts of karst forest were destroyed and subsequently colonized by nonnative invasives such as *Leucaena leucocephala* (Family: Fabaceae). The accidental introduction of the invasive brown treesnake in the 1940s has led to the extirpation or extinction of 10 of the 12 native forest birds on the island [44], reducing animal seed dispersal to negligible levels within the native forests [45,46]. In addition, invasive deer and pigs limit plant recruitment [47]. Collectively, these factors have led to the degradation of the native forest of the island [48].

The goal of forest restoration in the Marianas is to facilitate succession from degraded to native forest. Areas cleared of

**Figure 1.** Flowchart illustrating the steps of Spore. The model integrates both the characteristics of ecological function and function provider to identify key areas for rewilding. The landscape is divided into a user-defined grid. Cells are scored based on either their need of ecological function or capacity to host the function provider. A series of different scores allows to identify optimal areas for rewilding that can host the service provider while influencing the most areas in need of functional restoration. Finally, cells are grouped into management units using user-defined rules, which are scored and ranked. (Online version in colour.)

limestone forest and the karst substrate quickly get overcome by grasses and shrubs, then nonnative woody species (often dominated by *L. leucocephala*), which gradually transitions to a forest cover with mixed introduced and native species, culminating in the native-dominated forest. However, without seed dispersers present, succession and regeneration is slow to non-existent [46] and forests restoration requires manual seeding or outplanting of native species. To reach a self-regulating state, seed dispersal must be restored. With the extirpation of vertebrate seed dispersers from nearly all of the forests on the island, the only pathway available is rewilding through facilitated range expansion or reintroduction. The såli (Micronesian starling, *Aplonis opaca*) is the only native frugivorous bird still present on Guam, with a small population persisting within Andersen Air Force Base, a military installation in northern Guam. Among all birds historically present on Guam and still present in Micronesia, the såli presents the broadest diet (with Totot, the Mariana fruit dove [49]), the largest home range [50] and frequently crosses ecotones [50]. The såli frequently disperse seeds in the range of 100 m, but can disperse seeds more than 500 m in rare events [36]. Therefore, the såli is an ideal candidate for range expansion to restore the ecological function of seed dispersal. A pre-requisite for restoring ecological function through rewilding would be controlling invasive ungulates and snakes.

## (ii) Landscape evaluation

Successful functional restoration through rewilding within a heterogeneous landscape starts with identifying areas where functional restoration could benefit the landscape and areas where habitat is suitable for the chosen function provider. In our case study, we divided Guam into a 30 × 30 m grid. This is associated with the rasterization of the landcover map to the same resolution. Resolution is user-defined, and thus flexible. When defining resolution, we encourage users to follow key guidelines:

— Select a cell resolution that conserves the spatial information provided by the landcover map; a patchy and heterogeneous landscape will require higher spatial resolution than a homogeneous landscape. The area assigned to each landcover in

the original and the rasterized map should be roughly equivalent; if it is not, then a finer resolution may be needed.
— Select a resolution that matches the spatial scales of function. Resolution should always be smaller than the spatial scale at which function is provided. The use of a higher resolution enables the function to be mapped at finer spatial scales.
— Select a resolution that allows for reasonable computing times. Selecting a high resolution will increase the number of entities in the model and thus increase computing time.

We encourage Spore users to explicitly describe how resolution was selected to allow for replicability. In our case study, we selected a 30 × 30 m grid because it has better representation of the land covers than a 50 m resolution, was at least 10 times smaller than the maximum distance function is considered (500 m dispersal of seeds by birds), and provided reasonable computing times compared to a 20 m resolution (which increased simulation times by a third).

*Priority score: where is ecological function needed?* To identify where the ecological function is needed across the landscape, Spore assigns a priority score to each cell based on expert opinion or empirical data. This priority score, normalized between 0 and 1, should reflect the level of function need and the benefit of functional restoration for each cell. For example, areas with high ecological memory [51], defined as 'an ecosystem's accumulated abiotic and biotic material and information legacies from past dynamics', should be prioritized over areas with lower ecological memory. At the community (or landcover) level, areas with more intact terrain and those that retain remnant populations or propagule sources of native species would be prioritized. Ultimately, active rewilding would contribute towards the return of these ecosystems to a self-sustaining state.

In our case study, we identified three forest types that require functional restoration of seed dispersal to avoid further degradation and have the potential to return to a self-sustaining state in the near future: intact limestone forest, degraded limestone forest and *Leucaena thicket*. Restoration of seed dispersal to other landcover types, such as grassland, scrub-shrub or developed areas, would be insufficient for forest recovery due to competition from grasses [52] and fires [53], both preventing

seedling establishment. Within these landcover types, we assigned priority scores that reflect the ecological memory of each. Intact limestone forest was assigned the highest possible priority score because it is home to the native fleshy-fruited trees that serve as seed sources for the areas needing restoration and it typically retains the karst terrain structure associated with limestone forest which facilitates native species recruitment. Degraded limestone forest and *Leucaena* thicket were, respectively, assigned a priority score of 0.8 and 0.6, based on a gradient of degradation and thus necessary intervention required to be restored to an intact state.

*Habitat suitability score: where can function providers persist?* Areas where a given function provider can establish and persist are identified in Spore by measuring the suitability of each individual cell within the landscape. Habitat can be scored either as a binary value, with 0 being non-suitable and 1 suitable, or as a score between 0 and 1. The threshold at which a habitat is considered suitable is user-defined and should reflect when reintroduction within this area would allow for the establishment of the species. For this step, we recommend using or adapting current habitat suitability indices available within the literature [54], following decision guidelines highlighted in such studies, and considering the following criteria:

— Could the species reach the site and disperse there without human assistance [55]? In many rewilding scenarios, natural colonization is impossible, and this criterion is met through active reintroductions [56].
— Are the abiotic environmental conditions eco-physiologically suitable for the species? This criterion integrates climatic conditions and resource availability.
— Is the biotic environment suitable for the species? Will the species compete with others or be subject to predation?

Sålis inhabit most habitat types on nearby islands that lack snakes [57,58], but on Guam are restricted to nesting and roosting in developed areas undergoing intensive snake control adjacent to native forest areas for foraging [59]. In this step of the model, we evaluate the habitat suitability without considering protection from snake predation, which is addressed in the management units section below. To create a habitat suitability score, each cell within the landscape is evaluated based on the cell's landcover and the amount of native forest within the potential home range (circle with a radius of 1083 m, based on H. S. Pollock *et al.* 2019, unpublished data). Cells are considered non-viable habitats if:

— The associated landcover is either bare rock, mangrove, open water or sand, which would lead to the absence of nesting areas within the cell.
— If there is less than 10% native forest within the home range, representing a minimum viable amount of native plant species as a food resource. This figure represents the approximate proportion of native forest within the home ranges of individuals within the remnant population on Andersen Air Force Base.

### (iii) Function score: where is functional restoration by the function provider possible?

For functions that require access to particular landcover types or topographical characteristics, and that operate across space (e.g. seed dispersal and pollination require visits to native plants; predation effects might be most important in riparian areas), characteristics associated with the function provider influence the potential area where restoration is possible. To identify areas where functional restoration is possible with a given function provider, Spore combines the priority score for each cell with function-specific characteristics of the function provider

to create a function score for that cell. We recommend normalizing this score between 0 and 1, and considering the following questions when developing the algorithm behind the score calculation:

— How does landscape heterogeneity influence the spatial distribution of the function? For example, is the function only found near a specific landcover type? Does it depend on topographical characteristics?
— How does the biology of the function provider influence the provision of the ecological function? Does habitat usage matter? Are there spatial limitations like distance that limit the function?

In Spore-Guam, each cell associated with one of the three prioritized landcover types is assigned a function score calculated following these criteria:

— To facilitate forest recovery, dispersers must move native seeds from intact to the degraded forest. Thus, both the density of intact forest within a given distance (as determined by the identity of the selected seed disperser) of the focal cell and the distance from the focal cell to the closest intact forest cell are considered when calculating the function score.
— We selected a distance-to- intact -forest threshold of 500 m based on the 99th percentile of såli dispersal distances [36].

### (iv) Rewilding score: selecting optimal locations for rewilding

Once we have identified where functional restoration could realistically occur and be beneficial, the next step is to combine the function score with the habitat suitability score to produce a rewilding score, which represents the ecological benefit of rewilding in that particular cell. A rewilding score is assigned to every cell that could potentially host the function provider (i.e. cells that are considered unsuitable are not assigned a rewilding score). This rewilding score can be adapted to integrate multiple function scores if needed. We recommend normalizing the score between 0 (low priority) and 1 (high priority), and considering the following questions when developing the algorithm behind the score calculation:

— How would ecological function be provided throughout the landscape if the provider was to be reintroduced within this cell? For example, does the provisioning of the function decrease with distance from the centre of the home range, or does function vary with landcover?

In Spore-Guam, the model selects all cells that are suitable habitats for the Såli and identifies the region around that cell which would constitute the bird's diurnal home range (1083 m, based on H. S. Pollock *et al.* 2019, unpublished data) if birds were to be reintroduced within the cell. The score is then calculated by summing up the function scores of all cells within the home range, weighted by distance to account for greater visitation to nearby cells. We use a linear function based on the distance to simulate this, ranging from a value of 1 for the scored cell, to 0 for cells located at the maximum home range distance (1083 m).

### (v) Management units: considering other societal factors important for rewilding success

While the rewilding score highlights optimal ecological outcomes, rewilding often involves active management in the short-term. To facilitate successful rewilding strategies, we group areas into entities that can be managed similarly. These management units are built by clustering neighbouring cells that share important, user-defined characteristics relevant to management. For example, potential conflicts between humans

and wildlife, which occur frequently around the globe [60], could be considered during this step by selecting areas far enough from urban habitats. Another important factor is land ownership, which plays a key role in the selection of conservation areas [61], as coordinating management across land owners is often difficult. We recommend using GIS software to create management units, since aggregation of individuals cells can be done using built-in functions. Questions to consider include:

— Which socio-ecological factors vary across space and affect management decisions? e.g. land ownership, landcover type
— Are there species interactions that involve different management strategies (i.e. control of non-native species or predators) in different habitats?
— Does the reintroduction and monitoring process vary based on landcover type?
— How does the presence or density of the human community affect management?

Once these rules are defined and the management units created, each unit is assigned a score, based on the rewilding values of the component cells. This score is used to rank management units and thus, identify the priority areas for rewilding to restore ecological function.

In this version of Spore on Guam, we identify management units based on the likely methods for controlling snakes, a necessity for såli persistence. Currently, forest restoration necessitates human intervention through perpetual manual seeding of native species. By restoring såli, and thus the seed dispersal process to these areas, gardening the forest would become unnecessary. Nevertheless, until snakes are eradicated from the island, snake management will be a necessary human intervention to allow for the restoration of extirpated vertebrates and their associated ecological functions. In undeveloped areas, snake control is likely to be accomplished by dropping toxicants attached to dead mice [62] into areas surrounded by an exclosure fence to prevent immigration [63]. Since toxicant drops are undesirable in developed areas, snake control is likely to be accomplished using bait tubes [64]. We do not consider land ownership in our example; however, we recognize that land ownership on Guam is perhaps the most important consideration in the conservation planning process given the contentious history of federal control of the land on the island. This will be included in future versions of Spore-Guam. The management units were assigned using ArcGIS Pro v. 2.2.0 and details of the process are available in the electronic supplementary material, appendix A.

## 3. Results

### (a) Priority score: identifying and prioritizing areas where functional restoration is needed

We identified three forest types that would benefit from the restoration of seed dispersal (figure 2a). The remaining intact limestone forest, the highest priority of the three forest types, is primarily located in the northern parts of the island on the coasts, with a small patch found in the southwest of the island. Mixed introduced forest covers most of the island and is considered a moderate priority (0.8) due to the need for non-native species control in combination with the restoration of seed dispersal. Finally, Leucaena thicket is found mostly all along the eastern coast, and is assigned the lowest priority score (0.6) due to the need for extensive management of the habitat in addition to seed dispersal restoration.

### (b) Habitat suitability score: identifying potential habitats for the ecological function provider

Among all landcovers, 11 had at least one cell that met the habitat requirements for såli (see electronic supplementary material, table B.1). The intact forest was the landcover type with the most surface that could host såli (98% of intact forest cells), representing 44% of the total såli habitat area (figure 2b). Developed land was second, with 19% of its cells being potential habitat, representing 17% of the total såli habitat area, followed by mixed introduced forest with 12% of its total cells representing 13.5% of the total såli habitat area.

### (c) Function score: identifying candidate areas for restoration of seed dispersal by the function provider

Thirty-seven per cent of the area assigned to the intact forest, mixed introduced forest, and Leucaena thicket received a positive function score and would benefit from the restoration of the ecological function, seed dispersal (figure 2c). All intact forest cells were assigned a positive function score, compared to only 14% of mixed introduced forest and 28% of Leucaena thicket cells (see electronic supplementary material, table B.2), mostly because these landcovers were too far from native forest to receive native seeds. Cells that could benefit from functional restoration represented 14.5% of the total island area. Across the island, function scores were highest in intact forest and declined with increasing distance from intact forest cells.

### (d) Rewilding score: identifying optimal areas for rewilding

The rewilding score was highest within intact limestone forests and slowly decreased with increasing distance from intact forest (figure 2d). Nearly all intact forest on the island could support the såli and would benefit from the recovery of ecological function. The total area covered by intact forest cells with a positive rewilding score was almost triple the area compared to any other landcover type (see electronic supplementary material, figure B.1). Mixed introduced forest, mixed grass, scrub shrub and developed are other landcovers for rewilding såli which cover a sufficient amount of the island (over 1%) and have high rewilding scores.

### (e) Identifying and ranking management units

We produced a map of all management units identified across the island (figure 2e) ranked by management score. We identified three regions with high potential for restoring ecological function through rewilding (figure 2f): the northern tip of the island which covers Uranao, Finegayan, Litekyan and Tarague, where Andersen Air Force based is located, combined with the Guam National Wildlife Refuge; Anao Conservation Area and adjacent lands along the northeast coast of the island; and the limestone forest within Fena located near the southwest coast of Guam. All three areas are shared between the military, the Government of Guam and private landowners. The largest existing såli population is in a developed area of Andersen Air Force Base, which did

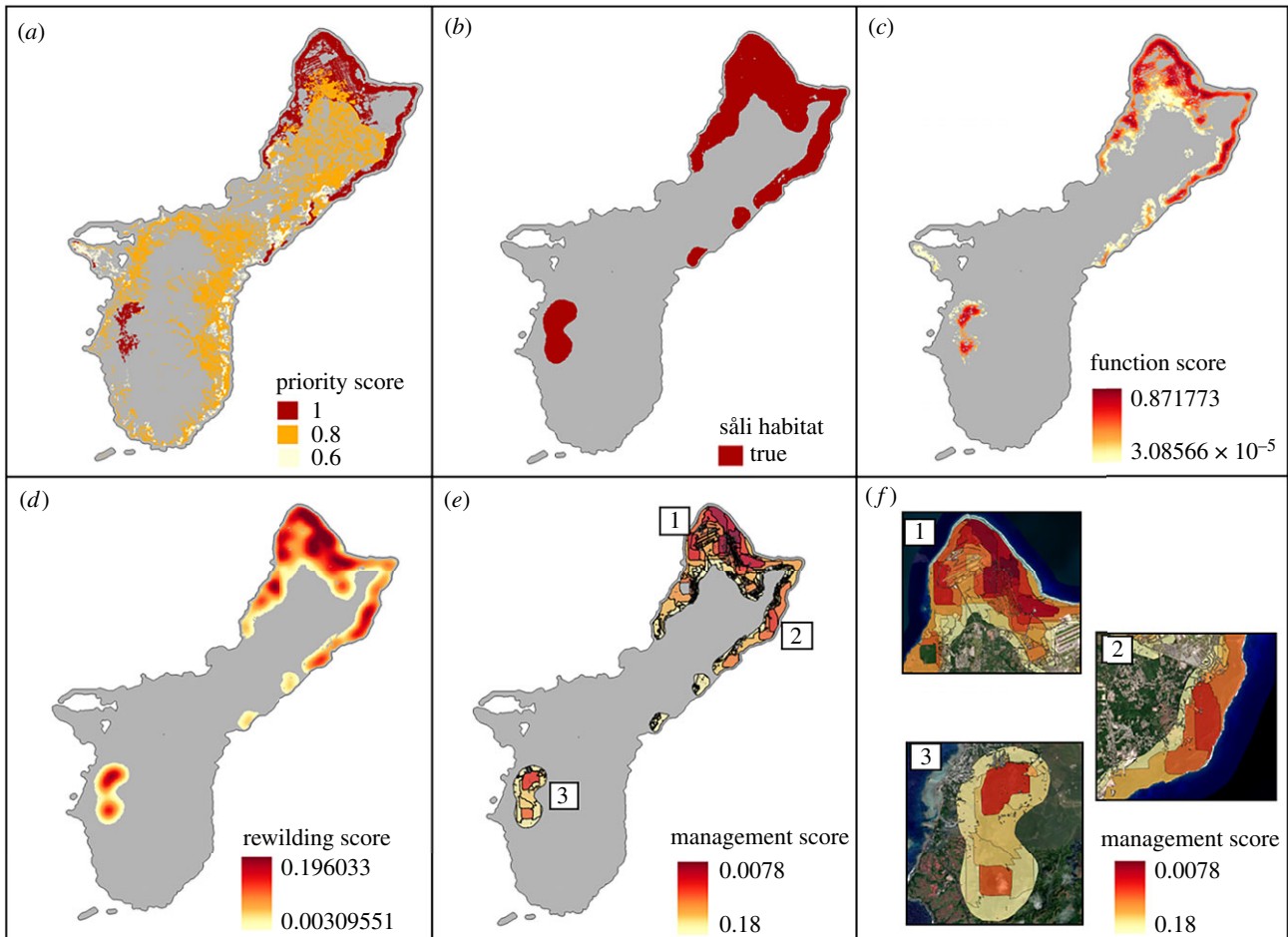

**Figure 2.** Spatial distribution across Guam of (*a*) areas where ecological function is needed and prioritized through a priority score, (*b*) the habitat suitability map for the såli, (*c*) areas where restoring seed dispersal by såli would be beneficial, represented by the function score, (*d*) areas where såli rewilding is likely to be successful and beneficial, represented by the rewilding score, and (*e*) the different management units identified throughout the island of Guam and ranked using the management score. Three general areas of interest can be highlighted throughout the island (*f*) with (1) the northern part of the island regrouping Uranao, Finegayan, Litekyan and Tarague, (2) the Anao cliff line, and (3) Fena. (Online version in colour.)

not show up as having a high rewilding score, but is adjacent to two regions of interest. Some såli individuals and pairs inhabiting urban areas on the island are not adjacent to areas with positive rewilding scores, indicating that a strategy focused on building up these populations is unlikely to provide needed ecological benefits of rewilding, and instead likely to lead to dispersal of non-native tree species.

## 4. Discussion

We developed a framework for mapping ecological function with habitat suitability of the function provider to identify optimal areas for restoring ecological function through rewilding. This approach is a valuable addition to the conservation manager's toolbox because it enables a more strategic approach to rewilding, extending beyond the needs of the focal species being rewilded, and contributing an ecological function perspective to a conservation approach typically driven by feasibility and economics. Our Guam case study identified three areas as prime sites for rewilding from an ecological perspective, where såli range expansion would maximize seed dispersal benefits. Current conservation efforts are already focused on some portions of these areas, but most of the limestone forest in two of the regions has received little conservation funding or attention. By matching the ecosystem

function to the abilities of the species under consideration for rewilding, and doing so across an entire landscape rather than only within areas already set aside for conservation, this approach may identify new areas to consider and stakeholders to engage in ecosystem rehabilitation efforts.

The identification of areas that could benefit from såli seed dispersal on Guam is driven primarily by the location of intact forest, which serves as a seed source for all other forest types. Since seed dispersers do not distinguish between native and non-native species [65], and sometimes even prefer non-native fruits [66], adding seed dispersers to parts of the island where såli could persist but that are far from the intact forest could be counter-productive. Såli would probably disperse non-native species due to a lack of native seed sources in their home range, perpetuating the dominance of introduced forest [67]. This highlights the importance of considering how interactions between landscape structure and animal behaviour affect the spatial distribution of ecological function [68].

The flexibility of the Spore framework allows for the exploration of a wide variety of case studies. Through simple adjustments of its scoring system, Spore can be applied to ecosystem functions such as pollination or pest predation, which play important roles in agricultural systems [69,70]. Spore is also capable of multi-function, multi-species and community-based simulations, integrating each component's

ecological requirements and biological characteristics. This would allow the study of tradeoffs between ecological functions and between the species that provide these functions. For example, additional candidates exist for restoring seed dispersal, including fanihi (Mariana fruit bat, *Pteropus mariannus*) and totot (Mariana fruit dove, *Ptilinopus roseicapilla*). These species differ in their habitat requirements and dispersal characteristics (home range size, diet, gut passage time), thus modelling how they could impact seed dispersal individually and collectively would help identify ideal community composition leading to greater efficiency of future rewilding scenarios. However, pushing for scenarios that favour certain species and habitats to enhance a function might also lead to risk–risk tradeoffs, where involuntary selection against other species and functions can happen [71]. For example, favouring the succession from degraded to intact forests on the nearby island of Saipan could lead to a decrease in nightingale reed-warbler populations, an endangered native bird that often uses the invasive tree *L. leucocephala* for nesting [72]. While it is difficult to consider all possible functions and interactions in ecosystem-level conservation approaches, using spatially explicit models that consider species interactions prior to making costly and impactful management decisions could help identify such tradeoffs [73].

The Spore framework could also be expanded to include a temporal component. Reintroductions should result in changes to the landscape, which in turn would affect both the priority score (where function is needed) and the habitat suitability score (where function providers can persist). An initial run of Spore could select areas for reintroduction based on optimal rewilding in the present-day landscape, then the input landcover map could be updated to reflect anticipated changes as a result of restoring ecological function on a location-relevant time scale. Then this new landcover map could be used the rerun Spore and select new priority rewilding areas for restoring dispersal or perhaps to consider additional function providers. For example, in our Guam case study mixed introduced forest with reintroduced seed dispersers would become an intact forest in the near future, and *Leucaena* forest would become mixed introduced. We could adjust our landcover map accordingly, then rerun Spore to identify how such changes might affect future management decisions, and thus create a step by step rewilding plan. The same approach could be taken to incorporate population dynamics of the service providers, by varying the way scores are calculated to reflect home range expansions or to select areas to prioritize to build corridors linking areas of interest. In scenarios where the areas needing the ecological function do not overlap with suitable habitat for the function providers, Spore could be used to promote natural (re)colonizations by identifying corridors between areas hosting existing populations of function providers and areas needing the given ecological function.

Finally, identifying and ranking management units is important for engaging stakeholders and informing management projects. Stakeholders more effectively engage in ecological problems with the use of dynamic maps [74]. Spore produces such outputs, tailored to serve as a decision-support tool that easily integrates feedback from stakeholders. While we consider feasibility constraints in the production of our management units, such as landcover type in relation to predator control, we currently produce rewilding scenarios from a theoretical and ecological point of view. However, many rewilding projects applied are executed by stakeholders with multiple goals and operational constraints [75]. Spore's flexibility in defining management units will allow stakeholders to reach compromises between the ecological optimum and operational feasibility. Other societal factors that influence decision-making could be added to Spore, including the addition of economic costs associated with management units and rewilding scenarios [76].

With rewilding projects increasing around the world [77,78], Spore is the first framework to combine habitat suitability for the reintroduction of species with the spatial distribution of ecological function at very fine spatial scales. Such a tool can improve the ecological outcome of rewilding, by maximizing the overlap between where the function is provided and where it is needed. By identifying priority areas and producing relevant maps, Spore can be used to actively engage stakeholders in a collaborative effort to construct effective rewilding projects.

Data accessibility. Code of the model are available on github at the following address: https://github.com/EBL-Marianas/Spore

Authors' contributions. H.T and H.R conceptualized the framework. H.T developed the model. H.T and H.R wrote the manuscript.

Competing interests. We declare we have no competing interests

Funding. This material is based upon work supported by the Strategic Environmental Research and Development Program and the US Army Corps of Engineers under Contract no. W912HQ16C0013 (Project RC-2441).

Acknowledgements. The authors would like to thank H. Pollock, M. Kastner and E. Rehm for their assistance with the såli data and their feedback on the paper, as well as both anonymous reviewers for their constructive feedback in helping us improve and broaden the scope of this framework. The authors would also like to thank all the members of our laboratory at Iowa State University for reading and providing comments on early drafts of this article.

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
