## [Reviewer comments · Proceedings of the Royal Society B: Biological Sciences]

Review History

RSPB-2019-1945.R0 (Original submission)

Review form: Reviewer 1

Recommendation

Major revision is needed (please make suggestions in comments)

Scientific importance: Is the manuscript an original and important contribution to its field?

Good

General interest: Is the paper of sufficient general interest?

Good

Quality of the paper: Is the overall quality of the paper suitable?

Good

Is the length of the paper justified?

Yes

Should the paper be seen by a specialist statistical reviewer?

No

Do you have any concerns about statistical analyses in this paper? If so, please specify them explicitly in your report.

Yes

It is a condition of publication that authors make their supporting data, code and materials available - either as supplementary material or hosted in an external repository. Please rate, if applicable, the supporting data on the following criteria.

Is it accessible?

No

Is it clear?

N/A

Is it adequate?

N/A

Do you have any ethical concerns with this paper?

No

Comments to the Author

Dear editor and authors,

Thierry and Rogers present a framework for planning rewilding actions. This conceptual model accommodates an ecological function within a landscape-filter/conditioning layer overlapped to a social pellicle, aiming to depict and predict the best places in a determined space to optimize the rewilding process. I would like to congratulate the authors for this approach. The proposal is entirely novel and provides an interesting applicable tool to management strategies focused on rewilding, once the real-world has become overwhelming defaunated. Yet – thinking beyond the mesocosm of the Guam Island and expecting broad applicability of the method, I have some suggestions/questions that can be addressed to increase the paper robustness and clarify some gaps, as follow:

Once the model is spatially explicit, space is discrete (gridded) and depends on the grid-size. Thus, the function score depends on the Sâli dispersal ability. My question refers to the model validation/calibration. For example, in Species Distribution Modelling (or Niche Modelling) the AUC value is a technique (even controversial) to depict the model accuracy. Another example of matrix-based statistic test to apply in a table derived from this grid-based model is the Patefield null models – under the null hypothesis of no association between rows and columns since your function score is spatially dependent (i.e., the grid neighborhood to a grid with high function score is prioritized). Further, even a way of simplifying validation by randomly distributing the values obtained for the function score within the full matrix/layer (considering a rectangular matrix). In summary, there are no evidence and methodology for model parameterization, calibration, and validation. Although I recognize that the Spore was developed based on ecological-induced criteria (according to your questions in section 2.1.2, 2.1.3, and 2.1.4), therefore generating a model suitable to rewilding the seed dispersal function, I suggest a model validation to allow the replication of Spore technique to other scales or to larger scales. Alternatively, discuss this issue.

L96: “The Spore framework supports rewilding projects that explore spatial distribution of ecological functions provided at fine spatial scales as well as the habitat suitability for rewilding species”. Does Spore “host” two or more functions? According to Fig. 1 seems so (concomitantly?), but according to the results the only function tested was dispersal seed. Have you ever wondered how the model would behave if instead of the crude function (i.e. seed dispersal) you replaced it with a functional diversity metric (that embodies more than one

function)? I understand that the function score can be a broad item, but also is appropriated to restore the functional diversity.

L104: “fine-scale spatial grid”. What is “fine-scale”? What the pixel size? Is flexible, can be changed to large-scale studies?

L117: “All code used can be found online at the following address: <https://github.com/EBL-Marianas/Spore>.” I consulted this repository from Sep-4 to Sep-16. During this period, I received a message as is “This depository is empty”. Thus, I do not are able to evaluate/understand the codes. Can you provide the codes in an Appendix?

L182. “Habitat can be scored either as a binary value, with 0 being non-suitable and 1 suitable, or as a score between 0 and 1.” Does this mean that the threshold does not necessarily have to be 0.5 to binary? For example, a score of 0.01 is suitable – even it is much closer to 0 than 1?

Minor issues:

L34: “...analogous species”. I would be careful about this sentence already in the summary. A decision maker may read only the summary and think that a non-native species may be a good option for rewilding. We know that most of the time it is not, with dramatic consequences for the local environment.

Fig. 1. In the data input section of the figure, the model indicates that can support more than one function. Yet, any ecological function can be addressed “one per time” in Spore leading to (perhaps) a misinterpretation.

Appendix A.

Function score

1) How was the priority coefficient threshold (e.g. 0.6 to *Leucaena* thicket) defined? Was it according to the “fruit/seed capacity” in each habitat type? Please include because this coefficient is weighting the score.

2) I must recognize that the authors encompass a time-dependent dynamic (dispersion) in the function score. That is good. The function score depends on two species traits: frugivory and dispersion ability (defined via home range, a powerful descriptor of the spatial requirements of an individual or population).

I hope that my suggestions will contribute to the improvement of the manuscript.

Review form: Reviewer 2

Recommendation

Major revision is needed (please make suggestions in comments)

Scientific importance: Is the manuscript an original and important contribution to its field?

Acceptable

General interest: Is the paper of sufficient general interest?

Good

Quality of the paper: Is the overall quality of the paper suitable?

Acceptable

Is the length of the paper justified?

Yes

Should the paper be seen by a specialist statistical reviewer?

No

Do you have any concerns about statistical analyses in this paper? If so, please specify them explicitly in your report.

No

It is a condition of publication that authors make their supporting data, code and materials available - either as supplementary material or hosted in an external repository. Please rate, if applicable, the supporting data on the following criteria.

Is it accessible?

No

Is it clear?

N/A

Is it adequate?

N/A

Do you have any ethical concerns with this paper?

No

Comments to the Author

The authors propose a framework that integrates habitat suitability mapping and function restoration potential in order to prioritize areas for (re)introduction and maximize the provision of functions. Although there are several points regarding the concepts and methods that need clarification, this is an interesting approach that is worth discussing. By addressing comments and questions, this manuscript and the approach developed by the authors has potential to better align (re)introductions with the rewilding principles, which can be useful for management and conservation planning.

General and major comments:

1. The premise that rewilding is only about the introduction of locally extinct or analogous species is not correct. There are different approaches to rewilding, one of them being trophic rewilding (sensu Svenning et al., 2016, PNAS, already cited in the MS) which puts more emphasis on introductions. However, rewilding is broader in terms of approaches and ecological processes to be considered and restored (Corlett, 2016 - already cited; Fernández et al., 2017; Jørgensen, 2014; Nogués-Bravo et al., 2016; Perino et al., 2019). In addition, the assumption that little consideration is given to where ecological function is most needed when rewilding (lines 35-36) is surprising, in light of most definitions of rewilding. Overall, the definition of rewilding, and link between (re)introductions, restored functions, and rewilding should be revised, and this would in my opinion, make the approach and its interest clearer. For instance, the first paragraph (lines 50 to 61) starts the introduction of the MS with reintroductions and jumps to the rewilding of the grey wolf (line 56) without explaining what rewilding is, and implying that rewilding here is a synonym of reintroducing (which it is not).
2. When discussing management and considering that rewilding aims at reducing gradually the human management of ecosystems, can you discuss how intense the human intervention should be for the reintroduction per se and subsequent management?
3. The SPORE framework is described as designed to evaluate spatially the need for a given ecological function and the suitability of the habitat for the function provider (Methods, lines 104-

106). But the Methods and Appendix call for some clarifications.

(1) In theory there shouldn't be a one-to-one relationship between the function and the provider, with several candidate species potentially providing a given function in the system. However, in the framework, the calculation for the function score seems tailored to the selected provider (e.g. with its dispersal ability). You do discuss in the Discussion the possibility to extend the framework to more species but this might come a bit too late.

(2) Furthermore, I find it difficult to understand how the "need for the given function", or Function score, is calculated in Appendix A. Are you identifying cells that cannot be reached for dispersal by the Sali, considering their current (limited) distribution and where there is a seed dispersing "gap"? Do you prioritize cells that are not occupied by other species that play the role of seed dispersers? Looking at the function score in Appendix A, I understand it as the potential for function restoration, if the Sali were to be reintroduced within its suitable habitat. But this is different from the need for function restoration.

4. Another point that is brought up in the discussion but that could be addressed sooner is the potential trade-offs between the functions being restored when selecting functions, and function providers in the model preparation (e.g. lines 124-127).

5. Would your framework allow to also prioritize areas for habitat restoration if you didn't find any (or too little) match between the need for a restored function and the habitat suitability of the candidate function provider to be reintroduced? If yes, that would be quite interesting to discuss in my opinion. You could also discuss the potential to restore connectivity between patches of habitats to facilitate natural (re)colonisations rather than introductions.

6. Considerations for potential human/wildlife conflicts are missing within the framework, including under "2.1.4 other societal factors". For instance, if the missing function was the top-down control of grazers and browsers to facilitate secondary successions, you'd probably consider (re)introducing top carnivores. This can be very limiting for the reintroduction potential, although most likely not in this context, and should be discussed.

Specific or minor comments:

7. You might want to consider discussing and citing the IUCN Guidelines for Reintroductions and Other Conservation Translocations.

8. Out of curiosity, could you add a layer to your framework that would consider the population dynamics? The framework could allow you to prioritize areas for initial reintroductions, assuming that the population size would grow, with potential range expansion, which could after x years reach the full potential for function provision.

9. In Figure 2, it could be useful to add the Land Cover map of Guam and the current distribution of the Sali (or in the Appendix).

10. When you present the different current management types in Guam (e.g. lines 282-286), can you discuss how their current management plans could be compatible with the rewilding definition/principles (or vice versa)?

11. Can you discuss whether by increasing the population of Sali on the island, there would be a risk of increasing the population of brown tree snakes and expanding their range as well?

12. Two potentially additional references are Donlan et al., 2006 and Torres et al., 2018.

Suggested references

Corlett, R.T., 2016. Restoration, Reintroduction, and Rewilding in a Changing World. *Trends Ecol. Evol.* 2078.

Donlan, C.J., Berger, J., Bock, C.E., Bock, J.H., Burney, D.A., Estes, J.A., Foreman, D., Martin, P.S., Roemer, G.W., Smith, F.A., others, 2006. Pleistocene rewilding: an optimistic agenda for twenty-

first century conservation. *Am. Nat.* 168, 660–681.

Fernández, N., Navarro, L.M., Pereira, H.M., 2017. Rewilding: A Call for Boosting Ecological Complexity in Conservation. *Conserv. Lett.* 10, 276–278.

Jørgensen, D., 2014. Rethinking rewilding. *Geoforum*.

Nogués-Bravo, D., Simberloff, D., Rahbek, C., Sanders, N.J., 2016. Rewilding is the new Pandora's box in conservation. *Curr. Biol.* 26, R87–R91. <https://doi.org/10.1016/j.cub.2015.12.044>

Perino, A., Pereira, H.M., Navarro, L.M., Fernández, N., Bullock, J.M., Ceaușu, S., Cortés-Avizanda, A., Klink, R. van, Kuemmerle, T., Lomba, A., Pe'er, G., Pliening, T., Benayas, J.M.R., Sandom, C.J., Svenning, J.-C., Wheeler, H.C., 2019. Rewilding complex ecosystems. *Science* 364, eaav5570. <https://doi.org/10.1126/science.aav5570>

Torres, A., Fernández, N., zu Ermgassen, S., Helmer, W., Revilla, E., Saavedra, D., Perino, A., Mimet, A., Rey-Benayas, J.M., Selva, N., Schepers, F., Svenning, J.-C., Pereira, H.M., 2018. Measuring rewilding progress. *Philos. Trans. R. Soc. B Biol. Sci.* 373, 20170433. <https://doi.org/10.1098/rstb.2017.0433>

Decision letter (RSPB-2019-1945.R0)

17-Oct-2019

Dear Dr Thierry:

I am writing to inform you that your manuscript RSPB-2019-1945 entitled "Where to rewild? A conceptual framework to spatially optimize ecological function" has, in its current form, been rejected for publication in *Proceedings B*.

This action has been taken on the advice of referees, who have recommended that substantial revisions are necessary. With this in mind we would be happy to consider a resubmission, provided the comments of the referees are fully addressed. However please note that this is not a provisional acceptance.

The two reviewers have some substantial reservations about the paper that will require major revisions if the paper might become acceptable. These constructive critiques are numerous and you will need to address them individually and fully in a Response if resubmitting.

To upload a resubmitted manuscript, log into <http://mc.manuscriptcentral.com/prsb> and enter your Author Centre, where you will find your manuscript title listed under "Manuscripts with

Decisions." Under "Actions," click on "Create a Resubmission." Please be sure to indicate in your cover letter that it is a resubmission, and supply the previous reference number.

Please note that this decision may (or may not) have taken into account confidential comments.

In your revision process, please take a second look at how open your science is; our policy is that all data involved with the study should be made openly accessible-- see: <https://royalsociety.org/journals/ethics-policies/data-sharing-mining/> Insufficient sharing of data can delay or even cause rejection of a paper. Both reviewers noted that the GitHub code was not available. Please make it available for review.

Sincerely,
Professor John Hutchinson, Editor
mailto:proceedingsb@royalsociety.org

Reviewer(s)' Comments to Author:

Referee: 1

Comments to the Author(s)

Dear editor and authors,

Thierry and Rogers present a framework for planning rewilding actions. This conceptual model accommodates an ecological function within a landscape-filter/conditioning layer overlapped to a social pellicle, aiming to depict and predict the best places in a determined space to optimize the rewilding process. I would like to congratulate the authors for this approach. The proposal is entirely novel and provides an interesting applicable tool to management strategies focused on rewilding, once the real-world has become overwhelming defaunated. Yet – thinking beyond the mesocosm of the Guam Island and expecting broad applicability of the method, I have some suggestions/questions that can be addressed to increase the paper robustness and clarify some gaps, as follow:

Once the model is spatially explicit, space is discrete (gridded) and depends on the grid-size. Thus, the function score depends on the Sáli dispersal ability. My question refers to the model validation/calibration. For example, in Species Distribution Modelling (or Niche Modelling) the AUC value is a technique (even controversial) to depict the model accuracy. Another example of matrix-based statistic test to apply in a table derived from this grid-based model is the Patefield null models – under the null hypothesis of no association between rows and columns since your function score is spatially dependent (i.e., the grid neighborhood to a grid with high function score is prioritized). Further, even a way of simplifying validation by randomly distributing the values obtained for the function score within the full matrix/layer (considering a rectangular matrix). In summary, there are no evidence and methodology for model parameterization, calibration, and validation. Although I recognize that the Spore was developed based on ecological-induced criteria (according to your questions in section 2.1.2, 2.1.3, and 2.1.4), therefore generating a model suitable to rewilding the seed dispersal function, I suggest a model validation to allow the replication of Spore technique to other scales or to larger scales. Alternatively, discuss this issue.

L96: "The Spore framework supports rewilding projects that explore spatial distribution of ecological functions provided at fine spatial scales as well as the habitat suitability for rewilding species". Does Spore "host" two or more functions? According to Fig. 1 seems so (concomitantly?), but according to the results the only function tested was dispersal seed. Have you ever wondered how the model would behave if instead of the crude function (i.e. seed dispersal) you replaced it with a functional diversity metric (that embodies more than one

function)? I understand that the function score can be a broad item, but also is appropriated to restore the functional diversity.

L104: “fine-scale spatial grid”. What is “fine-scale”? What the pixel size? Is flexible, can be changed to large-scale studies?

L117: “All code used can be found online at the following address: <https://github.com/EBL-Marianas/Spore>.” I consulted this repository from Sep-4 to Sep-16. During this period, I received a message as is “This depository is empty”. Thus, I do not are able to evaluate/understand the codes. Can you provide the codes in an Appendix?

L182. “Habitat can be scored either as a binary value, with 0 being non-suitable and 1 suitable, or as a score between 0 and 1.” Does this mean that the threshold does not necessarily have to be 0.5 to binary? For example, a score of 0.01 is suitable – even it is much closer to 0 than 1?

Minor issues:

L34: “...analogous species”. I would be careful about this sentence already in the summary. A decision maker may read only the summary and think that a non-native species may be a good option for rewilding. We know that most of the time it is not, with dramatic consequences for the local environment.

Fig. 1. In the data input section of the figure, the model indicates that can support more than one function. Yet, any ecological function can be addressed “one per time” in Spore leading to (perhaps) a misinterpretation.

Appendix A.

Function score

- 1) How was the priority coefficient threshold (e.g. 0.6 to *Leucaena* thicket) defined? Was it according to the “fruit/seed capacity” in each habitat type? Please include because this coefficient is weighting the score.
- 2) I must recognize that the authors encompass a time-dependent dynamic (dispersion) in the function score. That is good. The function score depends on two species traits: frugivory and dispersion ability (defined via home range, a powerful descriptor of the spatial requirements of an individual or population).

I hope that my suggestions will contribute to the improvement of the manuscript.

Referee: 2

Comments to the Author(s)

The authors propose a framework that integrates habitat suitability mapping and function restoration potential in order to prioritize areas for (re)introduction and maximize the provision of functions. Although there are several points regarding the concepts and methods that need clarification, this is an interesting approach that is worth discussing. By addressing comments and questions, this manuscript and the approach developed by the authors has potential to better align (re)introductions with the rewilding principles, which can be useful for management and conservation planning.

General and major comments:

1. The premise that rewilding is only about the introduction of locally extinct or analogous species is not correct. There are different approaches to rewilding, one of them being trophic rewilding (*sensu* Svenning et al., 2016, PNAS, already cited in the MS) which puts more emphasis on introductions. However, rewilding is broader in terms of approaches and ecological processes

to be considered and restored (Corlett, 2016 - already cited; Fernández et al., 2017; Jørgensen, 2014; Nogués-Bravo et al., 2016; Perino et al., 2019). In addition, the assumption that little consideration is given to where ecological function is most needed when rewilding (lines 35-36) is surprising, in light of most definitions of rewilding. Overall, the definition of rewilding, and link between (re)introductions, restored functions, and rewilding should be revised, and this would in my opinion, make the approach and its interest clearer. For instance, the first paragraph (lines 50 to 61) starts the introduction of the MS with reintroductions and jumps to the rewilding of the grey wolf (line 56) without explaining what rewilding is, and implying that rewilding here is a synonym of reintroducing (which it is not).

2. When discussing management and considering that rewilding aims at reducing gradually the human management of ecosystems, can you discuss how intense the human intervention should be for the reintroduction per se and subsequent management?

3. The SPORE framework is described as designed to evaluate spatially the need for a given ecological function and the suitability of the habitat for the function provider (Methods, lines 104-106). But the Methods and Appendix call for some clarifications.

(1) In theory there shouldn't be a one-to-one relationship between the function and the provider, with several candidate species potentially providing a given function in the system. However, in the framework, the calculation for the function score seems tailored to the selected provider (e.g. with its dispersal ability). You do discuss in the Discussion the possibility to extend the framework to more species but this might come a bit too late.

(2) Furthermore, I find it difficult to understand how the "need for the given function", or Function score, is calculated in Appendix A. Are you identifying cells that cannot be reached for dispersal by the Sali, considering their current (limited) distribution and where there is a seed dispersing "gap"? Do you prioritize cells that are not occupied by other species that play the role of seed dispersers? Looking at the function score in Appendix A, I understand it as the potential for function restoration, if the Sali were to be reintroduced within its suitable habitat. But this is different from the need for function restoration.

4. Another point that is brought up in the discussion but that could be addressed sooner is the potential trade-offs between the functions being restored when selecting functions, and function providers in the model preparation (e.g. lines 124-127).

5. Would your framework allow to also prioritize areas for habitat restoration if you didn't find any (or too little) match between the need for a restored function and the habitat suitability of the candidate function provider to be reintroduced? If yes, that would be quite interesting to discuss in my opinion. You could also discuss the potential to restore connectivity between patches of habitats to facilitate natural (re)colonisations rather than introductions.

6. Considerations for potential human/wildlife conflicts are missing within the framework, including under "2.1.4 other societal factors". For instance, if the missing function was the top-down control of grazers and browsers to facilitate secondary successions, you'd probably consider (re)introducing top carnivores. This can be very limiting for the reintroduction potential, although most likely not in this context, and should be discussed.

Specific or minor comments:

7. You might want to consider discussing and citing the IUCN Guidelines for Reintroductions and Other Conservation Translocations.

8. Out of curiosity, could you add a layer to your framework that would consider the population dynamics? The framework could allow you to prioritize areas for initial reintroductions, assuming that the population size would grow, with potential range expansion, which could after x years reach the full potential for function provision.

9. In Figure 2, it could be useful to add the Land Cover map of Guam and the current distribution of the Sali (or in the Appendix).

10. When you present the different current management types in Guam (e.g. lines 282-286), can you discuss how their current management plans could be compatible with the rewilding definition/principles (or vice versa)?
11. Can you discuss whether by increasing the population of Sali on the island, there would be a risk of increasing the population of brown tree snakes and expanding their range as well?
12. Two potentially additional references are Donlan et al., 2006 and Torres et al., 2018.

Suggested references

- Corlett, R.T., 2016. Restoration, Reintroduction, and Rewilding in a Changing World. *Trends Ecol. Evol.* 2078.
- Donlan, C.J., Berger, J., Bock, C.E., Bock, J.H., Burney, D.A., Estes, J.A., Foreman, D., Martin, P.S., Roemer, G.W., Smith, F.A., others, 2006. Pleistocene rewilding: an optimistic agenda for twenty-first century conservation. *Am. Nat.* 168, 660–681.
- Fernández, N., Navarro, L.M., Pereira, H.M., 2017. Rewilding: A Call for Boosting Ecological Complexity in Conservation. *Conserv. Lett.* 10, 276–278.
- Jørgensen, D., 2014. Rethinking rewilding. *Geoforum*.
- Nogués-Bravo, D., Simberloff, D., Rahbek, C., Sanders, N.J., 2016. Rewilding is the new Pandora's box in conservation. *Curr. Biol.* 26, R87–R91. <https://doi.org/10.1016/j.cub.2015.12.044>
- Perino, A., Pereira, H.M., Navarro, L.M., Fernández, N., Bullock, J.M., Ceaușu, S., Cortés-Avizanda, A., Klink, R. van, Kuemmerle, T., Lomba, A., Pe'er, G., Pliening, T., Benayas, J.M.R., Sandom, C.J., Svenning, J.-C., Wheeler, H.C., 2019. Rewilding complex ecosystems. *Science* 364, eaav5570. <https://doi.org/10.1126/science.aav5570>
- Torres, A., Fernández, N., zu Ermgassen, S., Helmer, W., Revilla, E., Saavedra, D., Perino, A., Mimet, A., Rey-Benayas, J.M., Selva, N., Schepers, F., Svenning, J.-C., Pereira, H.M., 2018. Measuring rewilding progress. *Philos. Trans. R. Soc. B Biol. Sci.* 373, 20170433. <https://doi.org/10.1098/rstb.2017>.

Author's Response to Decision Letter for (RSPB-2019-1945.R0)

See Appendix A.

RSPB-2019-3017.R0

Review form: Reviewer 1

Recommendation

Accept as is

Scientific importance: Is the manuscript an original and important contribution to its field?

Good

General interest: Is the paper of sufficient general interest?

Good

Quality of the paper: Is the overall quality of the paper suitable?

Good

Is the length of the paper justified?

Yes

Should the paper be seen by a specialist statistical reviewer?

Yes

Do you have any concerns about statistical analyses in this paper? If so, please specify them explicitly in your report.

No

It is a condition of publication that authors make their supporting data, code and materials available - either as supplementary material or hosted in an external repository. Please rate, if applicable, the supporting data on the following criteria.

Is it accessible?

N/A

Is it clear?

N/A

Is it adequate?

N/A

Do you have any ethical concerns with this paper?

No

Comments to the Author

The authors presented a partially new version of the paper that addressed the majority of the issues raised by the review process. Significant changes were made in the Introduction, which in the current version clarified some gaps in the prior conceptual framework. Important changes also were made in the Methods section, solving the major points of misinterpretation, increasing details of model conception and leading to a better reading and interpretation flux. In doing so, some parts of the discussion were modified in accordance with the prior review recommendations.

Decision letter (RSPB-2019-3017.R0)

05-Feb-2020

Dear Dr Thierry

I am pleased to inform you that your Review manuscript RSPB-2019-3017 entitled "Where to rewild? A conceptual framework to spatially optimize ecological function" has been accepted for publication in Proceedings B. Congratulations!!

The referee(s) do not recommend any further changes. Therefore, please proof-read your manuscript carefully and upload your final files for publication. Because the schedule for publication is very tight, it is a condition of publication that you submit the revised version of your manuscript within 7 days. If you do not think you will be able to meet this date please let me know immediately.

To upload your manuscript, log into <http://mc.manuscriptcentral.com/prsb> and enter your Author Centre, where you will find your manuscript title listed under "Manuscripts with Decisions." Under "Actions," click on "Create a Revision." Your manuscript number has been appended to denote a revision.

You will be unable to make your revisions on the originally submitted version of the manuscript. Instead, upload a new version through your Author Centre.

- 1) A text file of the manuscript (doc, txt, rtf or tex), including the references, tables (including captions) and figure captions. Please remove any tracked changes from the text before submission. PDF files are not an accepted format for the "Main Document".
- 2) A separate electronic file of each figure (tiff, EPS or print-quality PDF preferred). The format should be produced directly from original creation package, or original software format. Please note that PowerPoint files are not accepted.

3) Electronic supplementary material: this should be contained in a separate file from the main text and the file name should contain the author's name and journal name, e.g. `authorname_procb_ESM_figures.pdf`

All supplementary materials accompanying an accepted article will be treated as in their final form. They will be published alongside the paper on the journal website and posted on the online figshare repository. Files on figshare will be made available approximately one week before the accompanying article so that the supplementary material can be attributed a unique DOI. Please see: <https://royalsociety.org/journals/authors/author-guidelines/>

4) Data-Sharing and data citation

It is a condition of publication that data supporting your paper are made available. Data should be made available either in the electronic supplementary material or through an appropriate repository. Details of how to access data should be included in your paper. Please see <https://royalsociety.org/journals/ethics-policies/data-sharing-mining/> for more details.

<http://datadryad.org/submit?journalID=RSPB&manu=RSPB-2019-3017> which will take you to your unique entry in the Dryad repository.

Once again, thank you for submitting your manuscript to Proceedings B and I look forward to receiving your final version. If you have any questions at all, please do not hesitate to get in touch.

Sincerely,
Professor John Hutchinson, Editor
<mailto:proceedingsb@royalsociety.org>

Reviewer(s)' Comments to Author:

Referee: 1

Comments to the Author(s).

The authors presented a partially new version of the paper that addressed the majority of the issues raised by the review process. Significant changes were made in the Introduction, which in

the current version clarified some gaps in the prior conceptual framework. Important changes also were made in the Methods section, solving the major points of misinterpretation, increasing details of model conception and leading to a better reading and interpretation flux. In doing so, some parts of the discussion were modified in accordance with the prior review recommendations.

Decision letter (RSPB-2019-3017.R1)

11-Feb-2020

Dear Dr Thierry

I am pleased to inform you that your manuscript entitled "Where to rewild? A conceptual framework to spatially optimize ecological function" has been accepted for publication in Proceedings B.

Open Access

Paper charges

Sincerely,
Proceedings B
<mailto:proceedingsb@royalsociety.org>

Appendix A

We would like to thank both anonymous reviewers for their excellent feedback that has helped us not only improve our framework, but also broaden the perspective of applications of it. A major part of the introduction was rewritten to include broader concepts of rewilding. We also revised the different scores in the framework to allow more flexibility in regard to key concepts suggested by the reviewers. All line references cited are based on the untracked version of the manuscript

Referee: 1

Thierry and Rogers present a framework for planning rewilding actions. This conceptual model accommodates an ecological function within a landscape-filter/conditioning layer overlapped to a social pellicle, aiming to depict and predict the best places in a determined space to optimize the rewilding process. I would like to congratulate the authors for this approach. The proposal is entirely novel and provides an interesting applicable tool to management strategies focused on rewilding, once the real-world has become overwhelming defaunated.

Thank you!

Yet—thinking beyond the mesocosm of the Guam Island and expecting broad applicability of the method, I have some suggestions/questions that can be addressed to increase the paper robustness and clarify some gaps, as follow:

1- Once the model is spatially explicit, space is discrete (gridded) and depends on the grid-size. Thus, the function score depends on the Sâli dispersal ability. My question refers to the model validation/calibration. For example, in Species Distribution Modelling (or Niche Modelling) the AUC value is a technique (even controversial) to depict the model accuracy. Another example of matrix-based statistic test to apply in a table derived from this grid-based model is the Patefield null models—under the null hypothesis of no association between rows and columns since your function score is spatially dependent (i.e., the grid neighborhood to a grid with high function score is prioritized). Further, even a way of simplifying validation by randomly distributing the values obtained for the function score within the full matrix/layer (considering a rectangular matrix). In summary, there are no evidence and methodology for model parameterization, calibration, and validation. Although I recognize that the Spore was developed based on ecological-induced criteria (according to your questions in section 2.1.2, 2.1.3, and 2.1.4), therefore generating a model suitable to rewilding the seed dispersal function, I suggest a model validation to allow the replication of Spore technique to other scales or to larger scales. Alternatively, discuss this issue.

We thank the reviewer for suggesting approaches that will allow greater adoption of this method to other processes and scales. We agree that model parameterization and calibration/validation are important components to most models. We believe we provided evidence and methodology for model parameterization (Throughout methods, but specifically L168-186 for spatial resolution and L188-207 for landscape prioritization), but agree that we did not incorporate model calibration or validation into our approach. While we appreciate the suggestions for validation approaches, it is not clear to us that these would be applicable to our particular model. The rules we used to define the habitat suitability score are based primarily on landcover types, as a result of previous research (e.g. birds can live in native forest). The rules used for the function score integrate the behavior of the function provider (e.g. Sâli can function as dispersers up to 500 m) and the assigned priority of each cell (e.g. native forest is prioritized) to determine where functional restoration by the provider is possible. It is not clear to us how these steps could be calibrated or

validated as the data used to parameterize the model was either a crude measure (binary values for suitable & unsuitable habitat) or data collected in another system (function score). We could randomly distribute the function scores across the island to assess how much the patchiness of the real landscape or the scale at which the function provider operates affects the findings (akin to the reviewer's suggestion above), but that would be answering a separate question, rather than validating this model. The grid resolution selected by the user is important, so we have added recommendations on selecting the appropriate grid resolution in section 2.1.2. We believe that the flexibility and transparency of the functions used to calculate scores provides reproducibility and transferability to new applications.

L96: "The Spore framework supports rewilding projects that explore spatial distribution of ecological functions provided at fine spatial scales as well as the habitat suitability for rewilding species". Does Spore "host" two or more functions? According to Fig. 1 seems so (concomitantly?), but according to the results the only function tested was dispersal seed. Have you ever wondered how the model would behave if instead of the crude function (i.e. seed dispersal) you replaced it with a functional diversity metric (that embodies more than one function)? I understand that the function score can be a broad item, but also is appropriated to restore the functional diversity.

In this example, we did indeed focus on a specific function. Functional diversity could be modelled as a score, since the function score is user-defined, but it would need to be linked to a function provider. It is not obvious to us how a single metric to represent multiple functions could be linked to a single function provider. The strength of Spore in to be able to breakdown the individual contribution of each species to each key function. Functional diversity if considered, could be integrated in the rewilding score to help select areas where multiple beneficial outcomes overlap. We have underlined this possibility to include an ensemble of functions in the description of function score and explore the possibility of considering several functions in the discussion.

Added sentences L113-118

L104: "fine-scale spatial grid". What is "fine-scale"? What the pixel size? Is flexible, can be changed to large-scale studies?

The resolution we used is a 30x30 meter grid. We added this information and some guidelines to help users when defining resolution.

Added sentences starting L 168

L117: "All code used can be found online at the following address: <https://github.com/EBL-Marianas/Spore>." I consulted this repository from Sep-4 to Sep-16. During this period, I received a message as is "This depository is empty". Thus, I do not are able to evaluate/understand the codes. Can you provide the codes in an Appendix?

We apologize for this omission. We have updated the github repository, so the code should now be accessible.

L182. "Habitat can be scored either as a binary value, with 0 being non-suitable and 1 suitable,

or as a score between 0 and 1.” Does this mean that the threshold does not necessarily have to be 0.5 to binary? For example, a score of 0.01 is suitable—even it is much closer to 0 than 1?

In our approach, suitability is scored as either 0 (Unsuitable) or 1 (Suitable). Other habitat suitability models provide scores that are continuous over a series of values. The threshold for defining a suitable habitat is user-defined and thus could be a cut-off at any value as long as it is ecologically justifiable.

Added this info more clearly in the methods section L209-215

Minor issues:

L34: “...analogous species”. I would be careful about this sentence already in the summary. A decision maker may read only the summary and think that a non-native species may be a good option for rewilding. We know that most of the time it is not, with dramatic consequences for the local environment.

We agree that there are significant risks of introducing non-native species. However, there are systems where native species are now extinct and the use of analogous species is the only option for restoring ecological function, such as the use of giant tortoises in Mauritius (Falcon and Hansen, 2018). We have removed “analogous species” from the abstract, but discuss it briefly in the introduction.

L54-56

Fig. 1. In the data input section of the figure, the model indicates that can support more than one function. Yet, any ecological function can be addressed “one per time” in Spore leading to (perhaps) a misinterpretation.

We’ve changed the conceptual figure and added an extra step to highlight that the rewilding score can integrate multiple function scores. While we focus on a simple example of one function and one species, complexity can be increased if needed. We discuss this more throughout the paper but specifically added mentions to this L113-118 and L263-264.

Appendix A.

Function score

1) How was the priority coefficient threshold (e.g. 0.6 to *Leucaena* thicket) defined? Was it according to the “fruit/seed capacity” in each habitat type? Please include because this coefficient is weighting the score.

Priority coefficients are selected based on the importance of restoring function to that area, which was determined by expert opinion. We took two primary factors into consideration: 1) contribution to maintaining a robust native seed source and 2) effort required, aside from restoration of seed dispersal, to restore the area to a native state. These values are akin to the concept of “ecological memory” or a system’s biotic and abiotic legacy (Schweiger et al 2019). In our system, the native forest is the primary seed source of native seeds, followed by mixed introduced, then *Leucaena*. Similarly, restoration of seed dispersal would have the greatest individual effect in native forest, followed by mixed introduced, then *Leucaena*. All three would need to have ungulates controlled

as well, but the abiotic environment is likely the most conducive for seedling regeneration in native forest.

We have added a priority score in the framework (Section 2.1.2.1.) to clarify this important step and explained the choices for Guam L 196-207.

2) I must recognize that the authors encompass a time-dependent dynamic (dispersion) in the function score. That is good. The function score depends on two species traits: frugivory and dispersion ability (defined via home range, a powerful descriptor of the spatial requirements of an individual or population).

Thank you!

I hope that my suggestions will contribute to the improvement of the manuscript.

Referee: 2

Comments to the Author(s)

The authors propose a framework that integrates habitat suitability mapping and function restoration potential in order to prioritize areas for (re)introduction and maximize the provision of functions. Although there are several points regarding the concepts and methods that need clarification, this is an interesting approach that is worth discussing. By addressing comments and questions, this manuscript and the approach developed by the authors has potential to better align (re)introductions with the rewilding principles, which can be useful for management and conservation planning.

Thank you!

General and major comments:

1. The premise that rewilding is only about the introduction of locally extinct or analogous species is not correct. There are different approaches to rewilding, one of them being trophic rewilding (*sensu* Svenning et al., 2016, PNAS, already cited in the MS) which puts more emphasis on introductions. However, rewilding is broader in terms of approaches and ecological processes to be considered and restored (Corlett, 2016 - already cited; Fernández et al., 2017; Jørgensen, 2014; Nogués-Bravo et al., 2016; Perino et al., 2019). In addition, the assumption that little consideration is given to where ecological function is most needed when rewilding (lines 35-36) is surprising, in light of most definitions of rewilding. Overall, the definition of rewilding, and link between (re)introductions, restored functions, and rewilding should be revised, and this would in my opinion, make the approach and its interest clearer. For instance, the first paragraph (lines 50 to 61) starts the introduction of the MS with reintroductions and jumps to the rewilding of the grey wolf (line 56) without explaining what rewilding is, and implying that rewilding here is a synonym of reintroducing (which it is not).

Thank you for this helpful comment. There is an active debate in the literature over the term "rewilding", so we agree that it is important to faithfully reflect that complexity. We have rephrased the start of the introduction to encompass the different approaches to rewilding. We also recognize that function has long been a theoretical goal of rewilding, but maintain that the spatial

distribution of function needs is often a secondary consideration in rewilding efforts. We hope that our revised introduction more accurately reflects the state of the field today.

We have modified the introduction accordingly and added citations throughout.

2. When discussing management and considering that rewilding aims at reducing gradually the human management of ecosystems, can you discuss how intense the human intervention should be for the reintroduction per se and subsequent management?

We have added some guidelines highlighting the place of human intervention in the methods, both broadly and applied to our case study of Guam. Intervention effort is reflected through the priority score we added in the framework, which was part of the function score but clearly deserves to be discussed on its own. We also discussed human intervention linked to rewilding on Guam, where people are currently acting as the substitute seed dispersers in large-scale restoration projects. Expanding the range of Sali will require the control of snakes, but snake control provides a suite of benefits in addition to the ability to rewild Sali. Especially on islands, where invasive species have disproportionately large effects, persistent human intervention may be required over the long term (Hayward et al., 2019).

Added section 2.1.2.1 and modified section 2.1.5 to highlight this.

3. The SPORE framework is described as designed to evaluate spatially the need for a given ecological function and the suitability of the habitat for the function provider (Methods, lines 104-106). But the Methods and Appendix call for some clarifications.

(1) In theory there shouldn't be a one-to-one relationship between the function and the provider, with several candidate species potentially providing a given function in the system. However, in the framework, the calculation for the function score seems tailored to the selected provider (e.g. with its dispersal ability). You do discuss in the Discussion the possibility to extend the framework to more species but this might come a bit too late.

Our case study is indeed based on the relationship between a specific function and an ideal candidate selected through previous studies. However, it is possible to compare multiple function providers, and we have added a comment on this in the beginning of the methods as well as adjusted our conceptual diagram (Figure 1) to highlight this ability to handle increased complexity.

L113-118

(2) Furthermore, I find it difficult to understand how the "need for the given function", or Function score, is calculated in Appendix A. Are you identifying cells that cannot be reached for dispersal by the Sali, considering their current (limited) distribution and where there is a seed dispersing "gap"? Do you prioritize cells that are not occupied by other species that play the role of seed dispersers? Looking at the function score in Appendix A, I understand it as the potential for function restoration, if the Sali were to be reintroduced within its suitable habitat. But this is different from the need for function restoration.

We have adjusted the description of this approach in the methods to clear up this confusion. We separated the priority score from the function score, with the priority score identifying areas where function is needed (in this case, all intact, degraded, and tangantangan forest are assumed to

need seed dispersal). Then, in the function score, we restrict the priority areas to the areas where Sáli could actually provide that function (i.e. those within 500 m of native forest, which is a conservative estimate of how far Sáli can move seeds). The same rewilding score map would be produced if we omitted this step since cells further than 500 m would receive a 0 when the rewilding score is calculated, but the function score step dramatically reduces overall computing time.

Added the priority score in the framework to indeed highlight this important part of the model.

4. Another point that is brought up in the discussion but that could be addressed sooner is the potential trade-offs between the functions being restored when selecting functions, and function providers in the model preparation (e.g. lines 124-127).

Indeed. We now introduce this concept early on in the methods. L113-118.

5. Would your framework allow to also prioritize areas for habitat restoration if you didn't find any (or too little) match between the need for a restored function and the habitat suitability of the candidate function provider to be reintroduced? If yes, that would be quite interesting to discuss in my opinion. You could also discuss the potential to restore connectivity between patches of habitats to facilitate natural (re)colonisations rather than introductions.

We have added these interesting suggestions to the discussion, as prioritization and consideration of corridors can easily be considered using the function score.

L402-418

6. Considerations for potential human/wildlife conflicts are missing within the framework, including under "2.1.4 other societal factors". For instance, if the missing function was the top-down control of grazers and browsers to facilitate secondary successions, you'd probably consider (re)introducing top carnivores. This can be very limiting for the reintroduction potential, although most likely not in this context, and should be discussed.

Human-wildlife conflict is a serious issue in many areas, particularly for instances of trophic rewilding. To accommodate considerations of human-wildlife conflict, the function score calculation could incorporate the distance from developed areas. We added mention of human-wildlife conflicts in the 2.1.5 section, L279-281.

Specific or minor comments:

7. You might want to consider discussing and citing the IUCN Guidelines for Reintroductions and Other Conservation Translocations.

This is now cited in the introduction.

L56-57

8. Out of curiosity, could you add a layer to your framework that would consider the population dynamics? The framework could allow you to prioritize areas for initial

reintroductions, assuming that the population size would grow, with potential range expansion, which could after x years reach the full potential for function provision.

Yes, indeed! Population dynamics of the function provider could be included by adding a temporal component to the model, as describe in L402-418.

9. In Figure 2, it could be useful to add the Land Cover map of Guam and the current distribution of the Sali (or in the Appendix).

Added both figures to the Appendix

10. When you present the different current management types in Guam (e.g. lines 282-286), can you discuss how their current management plans could be compatible with the rewilding definition/principles (or vice versa)?

We have clarified the management requirements associated with rewilding in the text. Reintroductions would require human intervention until island-wide snake eradication is possible. Short of that, construction and maintenance of snake-proof fences combined with toxicant drops targeting snakes will be required to restore bird populations. However, the cost of not restoring bird populations is the slow degradation of native forest and loss of native seed sources if nothing is done, or the expenses associated with manual seed and seedling additions throughout intact and degraded forest areas. There is no easy answer in a heavily invaded and degraded system.

L295-308

11. Can you discuss whether by increasing the population of Sali on the island, there would be a risk of increasing the population of brown tree snakes and expanding their range as well?

The snake population currently spans the entire island, and is sustained primarily by abundant invasive small mammals as well as native and non-native skinks and geckos. Sali recovery requires reduced snake populations, therefore snake control is a pre-requisite and bringing back Sali should not affect island-wide snake populations. A bigger concern is the increase in small mammals likely to occur if snakes are controlled, but managers are planning for simultaneous snake and rodent control to prevent this.

12. Two potentially additional references are Donlan et al., 2006 and Torres et al., 2018.

Added both.

Suggested references

Corlett, R.T., 2016. Restoration, Reintroduction, and Rewilding in a Changing World. Trends Ecol. Evol. 2078.

Donlan, C.J., Berger, J., Bock, C.E., Bock, J.H., Burney, D.A., Estes, J.A., Foreman, D., Martin, P.S., Roemer, G.W., Smith, F.A., others, 2006. Pleistocene rewilding: an optimistic agenda for twenty-first century conservation. Am. Nat. 168, 660–681.

- Fernández, N., Navarro, L.M., Pereira, H.M., 2017. Rewilding: A Call for Boosting Ecological Complexity in Conservation. *Conserv. Lett.* 10, 276–278.
- Jørgensen, D., 2014. Rethinking rewilding. *Geoforum*.
- Nogués-Bravo, D., Simberloff, D., Rahbek, C., Sanders, N.J., 2016. Rewilding is the new Pandora's box in conservation. *Curr. Biol.* 26, R87–R91. <https://doi.org/10.1016/j.cub.2015.12.044>
- Perino, A., Pereira, H.M., Navarro, L.M., Fernández, N., Bullock, J.M., Ceaușu, S., Cortés-Avizanda, A., Klink, R. van, Kuemmerle, T., Lomba, A., Pe'er, G., Plieninger, T., Benayas, J.M.R., Sandom, C.J., Svenning, J.-C., Wheeler, H.C., 2019. Rewilding complex ecosystems. *Science* 364, eaav5570. <https://doi.org/10.1126/science.aav5570>
- Torres, A., Fernández, N., zu Ermgassen, S., Helmer, W., Revilla, E., Saavedra, D., Perino, A., Mimet, A., Rey-Benayas, J.M., Selva, N., Schepers, F., Svenning, J.-C., Pereira, H.M., 2018. Measuring rewilding progress. *Philos. Trans. R. Soc. B Biol. Sci.* 373, 20170433. <https://doi.org/10.1098/rstb.2017.0433>

Added the references among these that were missing